# FUN2SPEC: CODE CONTRACT SYNTHESIS AT SCALE

## ABSTRACT

We present FUN2SPEC, the first industrial-strength tool guiding LLMs to synthesize C++ specifications in first-order logic with quantifiers. FUN2SPEC's very high accuracy (85%) employs an automated validation procedure with error-driven oracle feedback, and can even generate the *strongest* code contract 60% of the time. Our approach significantly outperforms previous methods, achieving 20-25% higher validity on standard benchmarks. We applied FUN2SPEC to very large scale industrial C++ codebases containing many millions of lines of code and performed comprehensive manual validation to confirm the quality and utility of the specification. FUN2SPEC stands as the first effective code contract synthesis tool for real-world large-scale low-level C++ programs, advancing the state-of-the-art in automated software analysis using LLMs.

## 1 INTRODUCTION

Formal specifications provide essential descriptions for understanding and reasoning about programs (Gaudel, 1994). However, the significant manual effort required to write and maintain formal specifications in large codebases forces many large-scale projects to rely solely on natural language documentation to convey program intent. Specification synthesis—the problem of automatically inferring formal specifications from program implementations—offers a promising solution to this challenge. While specification synthesis is theoretically undecidable (like its dual problem, program synthesis), recent Large Language Models (LLMs) have demonstrated impressive capabilities in generating complex programs and reasoning about code semantics, making practical specification synthesis increasingly feasible. LLM-based agent systems have successfully addressed issues in large real-world codebases (Chen et al., 2021; Jimenez et al., 2024). This raises a compelling question: can LLM-based frameworks effectively synthesize formal specifications for large codebases at scale?

We present FUN2SPEC, a framework that generates formal specifications for complex, real-world codebases. While LLMs have shown promise in generating postconditions for small, independent problems (Endres et al., 2024b), real-world repositories present substantial challenges: complex interdependencies, diverse programming patterns, and extensive codebases where manual specification becomes impractical. FUN2SPEC addresses these challenges through an expressive specification language and a self-correcting refinement process. Our comprehensive evaluation not only demonstrates FUN2SPEC's effectiveness at scale but also includes extensive manual validation and qualitative analysis, confirming that these automatically generated specifications accurately capture program intent and provide practical utility for developers.

While recent code generation agents Anthropic (2024); Wang et al. (2025) have shown remarkable capabilities in writing and modifying code, formal specification synthesis addresses a fundamentally different challenge: understanding program semantics to generate logical postconditions rather than executable code. We envision future coding agents integrating automated specification synthesis to provide formal contracts for generated code, making our work a foundational step toward more rigorous AI-assisted development.

Our framework FUN2SPEC processes large C++ codebases through five interconnected components. First, our Code Miner extracts functions, type information, and unit tests from the repository. Next, we prompt the LLM with mined contextual information to generate specifications in our structured first-order logic syntax. We validate these specifications using a parser and translate them into executable C++ assertions. Our Specification Tester embeds these translated assertions into the codebase and executes existing unit tests to validate their semantic correctness. When parsing or

Figure 1: Workflow of FUN2SPEC: First, the code miner parses the repository to extract relevant context and tests for each function. Next, the LLM is prompted with context to infer the postcondition specification using CoT reasoning. The generated postcondition is then validated and translated by a parser into valid C++ expressions. If parsing or compilation fails, error summaries are fed back to the LLM in a self-correcting loop. Successfully validated postconditions are embedded into the function and validated through unit tests.

compilation errors occur, our system generates targeted error summaries and feeds them back to the LLM, creating a self-correcting refinement loop that improves specification quality.

**Contributions.** We make these significant contributions:
- We create FUN2SPEC, a framework that synthesizes formal specifications for large-scale C++ repositories through our AI pipeline (Mine, Generate, Parse, Test, and Refine) using a formal first-order logic syntax that includes quantifiers.
- We implement an effective parsing approach that validates and translates LLM-generated logical specifications into executable C++ assertions, with error-driven feedback that enables refinement.
- We demonstrate FUN2SPEC outperforms the state-of-the-art NL2POST (Endres et al., 2024a) approach by 20-35 percentage points in test validity on HumanEval and FormalSpecCPP benchmarks (Chen et al., 2021; Chakraborty et al., 2025) across multiple models.
- We comprehensively evaluate FUN2SPEC with SOTA LLMs on industrial-scale C++ projects containing millions of lines of code used daily by thousands of engineers, and conduct extensive manual validation to confirm specification quality.

The practical importance of FUN2SPEC is underscored by ongoing efforts to standardize C++ contracts within the language specification, with major compilers (GNU C v16, LLVM clang 22) planned to support native contracts by 2025, positioning FUN2SPEC's automated specification synthesis as a critical capability for the broader software engineering ecosystem.

## 2 PROBLEM STATEMENT

**Formal Specifications.** A specification for a program defines its intended behavior, describing what the program should do rather than how it achieves it. It is a formal contract between the program and its users, outlining the inputs, outputs, and expected behavior. For example, the intent of function *max*(int x, int y) is to return x if $x \geq x$ or $y$ otherwise.

A contract relates the values of the input program variables (pre-state of a program) to the values of the program variables and the output value of the function after it executes (post-state of a program). A *precondition* is a formula in formal logic whose literals are the program variables before the function starts. Similarly, a *postcondition* is a formula over the program variables that must hold true after a function or program completes their execution. Common choices for logics used in these formulas are propositional logic (consisting of variables, constants, arithmetic, logical and relational operators) or first-order predicate logic (which also includes existential and universal quantifiers). In FUN2SPEC, we use a first-order logic (FOL) formula to represent specifications.

**Definition 2.1** (First-Order Logic Formula). A first-order logic formula $\phi$ is defined recursively as:

$$\phi ::= Pr(t_1, \ldots, t_n) \mid \neg\phi \mid \phi \wedge \psi \mid \phi \vee \psi \mid \phi \rightarrow \psi \mid \forall x.\phi \mid \exists x.\phi$$

where $Pr$ is a predicate symbol applied to terms $t_i$, $\phi$ and $\psi$ are FOL formulas, $\neg, \wedge, \vee, \rightarrow$ are logical connectives, and $\forall, \exists$ are universal and existential quantifiers, respectively.

A precondition and postcondition are generally related using a semantic triple of the form $\langle P \rangle c \langle Q \rangle$ where $P$ stands for the precondition, $c$ is the program fragment under consideration, and $Q$ is the postcondition. As introduced in Hoare (1969), this triple is *valid* when for all pre-states satisfying $P$, once code fragment $c$ has executed and if its execution terminates, then all post-states will satisfy $Q$.

**Definition 2.2** (Postcondition inference). Given a code fragment $c$ and the predicate $P$ that is assumed to be the precondition for $c$, determine the postcondition $Q$ that yields a valid semantic triple $\langle P \rangle c \langle Q \rangle$.

Postcondition inference is the main feature of FUN2SPEC whose design and architecture are described in Section 3. Postconditions are typically expressed as logical statements, ensuring that the program adheres to its intended behavior. A C++ postcondition is shown in listing 1. The verification of postconditions presents significant challenges and is often undecidable. Due to this complexity and the notable absence of mature C++ tools specifically designed for automated postcondition verification, we employ test suites as a practical proxy for validating the correctness of our inferred postconditions.

While documentation in natural language can be ambiguous, postconditions enforce guarantees during execution, providing stronger assurances of program reliability and easier debugging.

**Large Language Models.** Autoregressive large language models (LLMs) are trained to predict the next token in a sequence given its preceding context. Recent works have shown they can effectively translate natural language into formal languages like programming code, mathematical expressions, or structured queries.

```cpp
// Sorts elements in ascending order
void sort(vector<int>& arr);
// Postcondition: For all adjacent elements
// i and i+1, it holds that arr[i] <= arr[i+1]
assert([&]() {
for (size_t i = 0; i < arr.size() - 1; i++)
  if (arr[i] > arr[i+1]) return false;
return true;
}());
```

Listing 1: Example of a postcondition in C++

Our work focuses on solving the postcondition inference problem for functions $f$ in large real-world repository $\mathcal{R}$ using an LLM.

## 3 FUN2SPEC

In this section, we describe the design of our solution FUN2SPEC for postcondition inference of functions in large C++ projects. FUN2SPEC's workflow has three main stages as shown in Figure 1: (1) mine the target code repository to extract contextual information; (2) synthesize candidate postconditions by prompting the LLM with this context; and (3) validate candidate postconditions through automated testing. First, the Code Miner parses the codebase to extract relevant function-level context, including type information and comments. This context is combined with few-shot examples to prompt the LLM, which generates postcondition specifications. The generated post-conditions are then translated into valid language expressions, temporarily embedded within the codebase, and validated by running the existing unit tests to ensure correctness. If there is an issue in parsing or compiling the LLM-generated postcondition, the model is reprompted with an error summary to refine its output. The next sections describe each of the FUN2SPEC components.

### 3.1 CODE MINER

Code Miner extracts functions and relevant function data from the source code of the program. We iterate through each file in the repository and obtain the abstract syntax-tree (AST) representation using the Tree-sitter library. The AST representation is used to extract the documentation for the function, which FUN2SPEC later uses to query the LLM. We use the Clang library to retrieve function return types and unit tests. Extracting tests is challenging as many functions are tested indirectly through transitive calls rather than direct unit tests. We address this by tracing call paths to identify all tests that exercise each function, regardless of call depth.

Let $\mathcal{F}$ denote the set of all functions in the repository, and $\mathcal{T}$ represent the set of all unit tests in the repository. For each $f \in \mathcal{F}$, the Code Miner extracts a mapping $U : \mathcal{F} \to \mathcal{P}(\mathcal{T})$ where $U(f)$ denotes the set of all unit tests corresponding to the function $f$. This mapping is used for validating the postconditions generated by FUN2SPEC.

Additionally, FUN2SPEC extracts the canonical return types of each function using Clang. In real-world repositories, top-level return types are often aliased using typedef or using directives. It is important to provide the LLM with the resolved, canonical type information to ensure understanding of the underlying types. Failure to do so leads to inferring candidates contracts that are not well-typed for the program of interest, and therefore should never be considered as candidate contracts.

## 3.2 LLM GENERATOR

In the LLM generator step, FUN2SPEC uses an LLM to synthesize postconditions for C++ functions. Recent research has shown that LLMs are in-context learners and providing a small number of input-output examples in the prompt significantly improve their overall accuracy Brown et al. (2020). Similarly, chain-of-thought (CoT) reasoning, which encourages the model to generate intermediate reasoning steps before arriving at a final answer, has been proven to be an effective prompting technique for LLMs Wei et al. (2023).

The prompt template used by FUN2SPEC includes a structured instruction specifying the expected syntax of the postcondition. FUN2SPEC uses the grammar of first-order logic to represent postconditions, where logical operators (such as conjunction, disjunction, implication, and quantifiers) are applied to atoms of C++ expressions as shown in Listing 2. While we only construct such logical formulae to represent postconditions, the same grammar could also be used for preconditions and loop invariant inference. Our postcondition syntax is similar to ACSL (ANSI/ISO C Specification Language) (Baudin et al.; Correnson et al.), which is a popular specification language for C programs.

```
postcondition: logical_expr
logical_expr  : implication
              | logical_term
              | quantifier_expr
implication: logical_term "==>" logical_expr
logical_term: cpp_expression
            | logical_term ("&&" | "||")
                logical_term
            | "!" logical_term
            | "(" logical_expr ")"

quantifier_expr: quantifier "(" CNAME "," expr
    "," logical_expr ")"

quantifier: "FORALL" | "EXISTS"

cpp_expr: /* Any valid C++ expression */
```

Listing 2: Grammar for postcondition expression

The inclusion of quantifiers (FORALL and EXISTS) significantly expands the expressiveness of our contract language, allowing FUN2SPEC to reason about properties that apply to collections of elements or ranges of values. For instance, a postcondition can now specify that all elements in an array meet certain criteria (FORALL(i, arr, arr[i] > 0)) or that at least one element satisfies a given condition (EXISTS(i, arr, arr[i] == target)).

The instructions for the syntax are followed by four few-shot examples that demonstrate CoT reasoning, and the desired postconditions corresponding to the examples. Each example introduces a specific type of logical operation to the model. The LLM output includes both the derived postcondition and the step-by-step reasoning process used to reach it. In addition to improving the accuracy the reasoning provides a form of explanability to the result through the logical steps leading to the synthesized postcondition. Appendix A.1 presents the full template for the prompt.

## 3.3 SPECIFICATION TESTER

We now introduce the specification tester, whose role in FUN2SPEC is to filter out invalid candidate contracts by instrumenting available built-in tests for the project. In the Specification Tester, we use the unit test mapping $U$, computed by the Code Miner, to evaluate the validity of the postcondition. A postcondition is deemed invalid if it fails to hold for even a single execution of a unit test. Since the program's intent is expressed only in natural language, it is not feasible to definitively evaluate the true validity of the postcondition. In practice, test-validity serves as an over-approximation of the post-condition validity, and will not guarantee contract correctness in itself.

**Formal Validity Measure.** We define the average test validity for set of functions $\mathcal{F}$ as:

**Definition 3.1** (Average Test Validity). If $Q_f$ denotes the postcondition for function $f$, $U(f)$ is the set of unit tests associated with $f$, as determined by the mapping $U$ and

$$\text{ATV}(\mathcal{F}) = \frac{1}{|\mathcal{F}|} \sum_{f \in \mathcal{F}} \prod_{t \in U(f)} t \vDash Q_f$$

where $t \vDash Q_f$ denotes that test $t$ semantically entails the specification $Q_f$, meaning the execution of test $t$ respects the constraints specified by $Q_f$ for all possible inputs and outputs. This evaluates to true (1) if test $t$ satisfies the specification $Q_f$ or false (0) otherwise. It is necessary for **all tests** to hold in $U(f)$ for the function $f$ and its inferred postcondition to contribute to the overall ATV.

**Postcondition Parsing and Translation.** We employ an Earley parser (Earley, 1970) to validate and translate the LLM-generated postconditions from our defined syntax into valid C++ assertions. The Earley parsing algorithm is particularly well-suited for this task as it can handle ambiguous grammars and provides detailed information about parsing failures.

The translation process converts logical constructs like implications, quantifiers, and logical operators into their C++ equivalents. For example, a quantifier expression `FORALL(i, arr, arr[i] > 0)` is translated into a C++ lambda expression that iteratively checks the condition for all elements in the array. This approach can be easily extended to other statically-typed programming languages by modifying the transformer component to generate assertions in the target language syntax, making FUN2SPEC adaptable to diverse development environments.

**Error Handling and Feedback Loop.** When parsing fails, a compilation error occurs, or a unit test fails to pass, we generate a succinct error summary as feedback to the LLM. This summary includes the specific error location and error type. Upon receiving such feedback, FUN2SPEC automatically reprompts the LLM with this error information, creating a feedback loop that allows the model to learn from its mistakes and regenerate an improved postcondition.

**Test Instrumentation.** Testing whether a postcondition holds on every execution of the function is a challenging problem. To address this issue, we parse the function $f$ and modify its implementation by replacing every occurrence of the return statement with a block of code that creates a temporary return value and evaluates the postcondition. This instrumentation ensures that the postcondition is checked at each possible exit point of the function, providing comprehensive validation across all execution paths exercised by the unit tests. For example, consider the function `divideArray` in Figure 2 that returns a pointer. The LLM generated postcondition (Listing 4) is transformed to a valid C++ expression (Listing 5). Next, a code block is inserted at every return statement to assign the return expression to a temporary variable and to evaluate the postcondition (Listing 6).

---

**Algorithm 1** FUN2SPEC Algorithm for Specification Synthesis

**Require:** Code repository $\mathcal{R}$, model $\mathcal{M}$, tests $\mathcal{T}$
**Ensure:** Average Test Validity Score ATV
1:  $\mathcal{F}, U \leftarrow \text{MINE}(\mathcal{R})$
2:  $ATV \leftarrow 0$
3:  **for** each function $f \in \mathcal{F}$ **do**
4:      $Q_{\text{valid}} \leftarrow \emptyset$
5:      $Q_{\text{cand}} \leftarrow \text{GENCANDIDATES}(\mathcal{M}, f)$
6:      success $\leftarrow$ false, attempts $\leftarrow$ max_attempts
7:      **while** !success **and** attempts $> 0$ **do**
8:          $Q' \leftarrow \text{PARSE}(Q_{\text{cand}})$
9:          **if** parse error **then**
10:             err $\leftarrow \text{ERRORSUMMARY}(\text{error})$
11:         **else**
12:             $f' \leftarrow \text{INSTRUMENT}(f, Q')$
13:             **if** compilation error **then**
14:                 err $\leftarrow \text{ERRORSUMMARY}(\text{error})$
15:             **else**
16:                 $V \leftarrow \text{TESTVALID}(f', U(f))$
17:                 **if** $V$ **then**
18:                     success $\leftarrow$ true
19:                     $Q_{\text{valid}} \leftarrow Q'$
20:                 **else**
21:                     err $\leftarrow \text{ERRORSUMMARY}(V)$
22:         **if** !success **then**
23:             $Q_{\text{cand}}$
24:             $\leftarrow \text{GENCANDIDATES}(\mathcal{M}, f, err)$
25:             attempts $\leftarrow$ attempts - 1
26:     **if** success **then**
27:         $ATV \leftarrow ATV + 1$
28:         $\text{CREATEPR}(f, Q_{\text{valid}})$
29:  return ATV/|F|

---

### 3.4 FUN2SPEC POSTCONDITION SYNTHESIS ALGORITHM

Algorithm 1 outlines our postcondition synthesis procedure and its connection to the ATV metric. Given a code repository $\mathcal{R}$, a language model $\mathcal{M}$, and unit tests $\mathcal{T}$, we begin by mining the repository to extract the function set $\mathcal{F}$ and their associated test mappings $U(f)$ (line 1). For each function $f \in \mathcal{F}$ (line 3), we invoke $\mathcal{M}$ to generate candidate postconditions (line 5). The algorithm then enters a validation feedback loop (lines 7-17) where each candidate postcondition undergoes multiple validation stages:

First, the Earley parser both validates the syntactic correctness of the candidate postcondition and transforms it from our formal syntax into valid C++ expressions (line 8). If parsing fails, an error

```
1  int* divideArray(const int* arr, int size,
2      int divider) {
3      if (divider == 0 ) {
4          return nullptr;
5      }
6      ...
7      if (arr == nullptr) {
8          return nullptr;
9      }
10     ...
11     int* out_arr = new int[size];
12     ...
13     return out_arr;
   }
```

Listing (3) Original Function Implementation

```
1  (arr == nullptr || divider == 0)
2      ==> res_tmp == nullptr
3  && (arr != nullptr || divider != 0)
4      ==> size(arr) == size(res_tmp)
```

Listing (4) LLM generated postcondition

```
1  (!(arr == nullptr || divider == 0)
2      || (res_tmp == nullptr))
3  && (!(arr != nullptr || divider!= 0)
4      || size(arr) == size(res_tmp))
```

Listing (5) Transformed postcondition to C++ syntax

```
1  int* divideArray(const int* arr, int size,
2      int divider) {
3      ...
4      int * res_tmp = out_arr;
5      assert(   /* From Listing 5: */
6          (!(arr == nullptr || divider == 0)
7              || res_tmp == nullptr)
8          && (!(arr != nullptr || divider!=0)
9              || size(arr) == size(res_tmp))
10     )
11     return res_tmp;
   }
```

Listing (6) Transformed Function Implementation for the final return expression in Listing 3 Line 12

Figure 2: Transformed function implementations for specification testing

summary is generated to provide detailed feedback. If parsing succeeds, the function is instrumented with the transformed C++ postcondition (line 11), where we insert assertions at every return point to validate the postcondition. This instrumentation may lead to compilation errors, which also generate error summaries for feedback to the language model.

If compilation succeeds, we execute the unit tests associated with $f$ to verify that the instrumented function satisfies the postcondition across all test cases (line 16). At each failure point (parsing or compilation), error summaries guide the language model to generate improved candidates (lines 23-24), creating a self-correcting loop until success or exhaustion. This final score (line 29) measures both our approach's effectiveness and specification quality.

## 4 EVALUATION

**Models.** We experiment on state-of-the-art open-weight LLMs, including Qwen (Qwen, 2024), Llama-3.1 (Llama, 2024), Gemma-2-9b-it (Abdin et al., 2024), and Phi-4 (Mesnard et al., 2024).

**Repositories.** We evaluate on HumanEval-CPP (Zheng et al., 2023) (a C++ translation of the original Python benchmark (Chen et al., 2021) with corresponding unit tests) and FormalSpecCPP (Chakraborty et al., 2025), a dataset containing C++ programs with well-defined ground truth preconditions and postconditions that are verified in

Table 1: Summary of repositories used in evaluation

| Repositories | BDE | BLAZINGMQ |
|---|---|---|
| Lines of Code | 3.4M | 727K |
| Functions w/ Tests | 3992 | 834 |
| Functions w/ Comments+Tests | 794 | 590 |

Dafny and manually validated on translation. Additionally, we consider two large open-source C++ repositories, BDE and BlazingMQ for the evaluation on large real-world repositories. BDE is a modular C++ library suite containing foundational components such as data structure algorithms and utilities used by thousands of developers. BLAZINGMQ is a high-performance, fault-tolerant message queue library used by thousands of low-latency applications. These projects are representative of common infrastructure libraries that have well-documented interfaces and strong test suites. Both analyzed projects are open-source GitHub projects and are heavily deployed within the technology industry. We automatically extract public functions with documentation and existing unit tests from both repositories as summarized in Table 1.

**Baseline.** We implement NL2POST Endres et al. (2024a) as a baseline, which translates natural language specifications into formal postconditions. The original code is not publicly available, so

we reimplement and adapt the algorithm to work with C++ while maintaining the same few-shot examples and hyperparameters as FUN2SPEC for fair comparison.

**Hyperparameters.** We use a prompt with four few-shot examples (Brown et al., 2020). The few-shot examples include CoT reasoning manually designed to show distinct types of postconditions. We use greedy decoding to sample the LLM output and set the maximum new tokens to 400 for standard instruct-tuned models and 800 tokens for reasoning models.

**Implementation.** We run experiments on a 48-core Intel Xeon Silver 4410Y CPU with one NVidia H100 GPU. FUN2SPEC is implemented using Hugging Face transformers library (Wolf et al., 2020) for LLM inference Clang and Tree-sitter for parsing the C++ code.

**Metric.** We use the following metrics to evaluate the quality of generated postconditions: (1) Test Valid (%): Postconditions that hold across all unit tests (Def. 3.1). (2) Test Invalid (%): Postconditions that fail on at least one test. (3) Compilation Error (%): Cases where the postcondition causes a compilation error or a timeout. (4) Invalid Formatting (%): LLM output is ill-formatted; no postcondition extracted. (5) Nontrivial (%): Postconditions that do not simplify to True. The model defaults to trivial (True) if unable to generate a valid one (e.g., "result != NULL || result == NULL"). (6) Avg. Atoms: Average number of atomic expressions in postconditions, indicating complexity.

## 4.1 BENCHMARK RESULTS

We evaluate FUN2SPEC against NL2POST Endres et al. (2024a) on two standard benchmarks: HumanEval-CPP and FormalSpecCPP. Table 2 presents the test-validity percentages across different models. We permit 1 refinement attempt and present results for scaling up the number of feedback iterations in Appendix A.4. On HumanEval-CPP, FUN2SPEC consistently outperforms NL2POST across all models, with improvements rang-

Table 2: Test-Validity (%) comparison between FUN2SPEC and NL2POST on benchmark datasets

| Model | HumanEval-CPP | | | FormalSpecCPP | | |
|---|---|---|---|---|---|---|
| | **FUN2SPEC** | **NL2POST** | $\Delta$ | **FUN2SPEC** | **NL2POST** | $\Delta$ |
| Qwen3-32B | 55.8 | 19.6 | +36.2 | 74.0 | 42.2 | +31.8 |
| Qwen2.5-32B | 63.2 | 20.2 | +43.0 | 76.0 | 43.1 | +32.9 |
| Qwen2.5-Coder-7B | 52.1 | 30.7 | +21.4 | 67.6 | 52.9 | +14.7 |
| Llama-3.1-8B | 17.2 | 8.0 | +9.2 | 20.0 | 30.4 | −10.4 |
| Gemma-2-9b-it | 36.2 | 14.1 | +22.1 | 59.0 | 37.3 | +21.7 |
| Phi-4 | 22.1 | 11.7 | +10.4 | 57.8 | 38.2 | +19.6 |
| Phi-4-mini | 25.2 | 16.6 | +8.6 | 43.1 | 28.4 | +14.7 |
| QwQ-32B | 6.1 | 3.1 | +3.0 | 48.0 | 11.8 | +36.2 |

ing from 8 to 43 percentage points and an average improvement of 19.2 points. The most notable gain is observed with Qwen3-32B, where FUN2SPEC achieves 55.8% test validity compared to NL2POST's 19.6%.

Similarly, on FormalSpecCPP, FUN2SPEC again shows stronger overall performance, particularly with larger models. Qwen2.5-32B-Instruct achieves the highest test validity at 76% with FUN2SPEC, compared to 43.1% with NL2POST. On average, FUN2SPEC improves test-validity by 20.1 percentage points over NL2POST on this benchmark. The improved performance of FUN2SPEC can be attributed to its feedback loop and systematic parsing approach, which allows it to refine postconditions.

We provide additional details on complexity of generated postconditions in Appendix A.2.1. We perform ablation in Appendix A.2 which shows that the standard setting with both reprompting and quantifier support consistently yields the best postcondition validity. For detailed comparison across all models and ablation settings, see Table 4.

## 4.2 POSTCONDITION GENERATION ON LARGE CODEBASES

Table 3 presents a comparative analysis of the performance of different models in generating postconditions for functions in repository BDE and BLAZINGMQ. The table shows varying performance across models. For instance, Qwen2.5-32B-Instruct achieves the highest rate of test-valid postconditions for both BDE (69.49%) and BLAZINGMQ (76.47%). When scaling the number of re-promting iterations from 1 to 10 (Appendix A.5), Qwen2.5-32B-Instruct achieves 86.94% test validity on BDE. In contrast, small models such as Phi-4-mini get relatively lower (33.44% and 20.13%) test-validity. We observe that majority of postconditions that are not test-valid are primarily due to compilation or

Table 3: Model Performance with FUN2SPEC on large C++ repositories

| Repo. | Model Name | Test Valid (%) | Test Invalid (%) | Compilation Error (%) | Formatting Error (%) | Trivial (%) | Avg. Atoms |
|-------|-----------|-----------|-----------|-----------|-----------|-----------|-----------|
| BDE | Qwen3-32B | 69.37 | 7.62 | 14.57 | 8.44 | 10.10 | 2.19 |
| | Qwen2.5-32B-Instruct | 69.41 | 6.09 | 14.80 | 9.70 | 14.97 | 2.29 |
| | Qwen2.5-Coder-7B-Instruct | 57.61 | 19.48 | 12.77 | 10.15 | 12.93 | 1.86 |
| | Llama-3.1-8B-Instruct | 8.01 | 67.65 | 20.10 | 4.25 | 0.65 | 1.61 |
| | Gemma-2-9b-it | 44.35 | 31.42 | 18.82 | 5.40 | 2.45 | 1.66 |
| | Phi-4 | 46.19 | 5.56 | 19.52 | 28.73 | 4.44 | 2.22 |
| | Phi-4-mini-instruct | 33.44 | 20.13 | 34.90 | 11.53 | 0.16 | 2.95 |
| | QwQ-32B | 12.36 | 0.57 | 5.89 | 81.18 | 1.15 | 1.44 |
| BLAZINGMQ | Qwen3-32B | 75.06 | 4.40 | 10.02 | 10.51 | 13.45 | 2.33 |
| | Qwen2.5-32B-Instruct | 73.00 | 3.52 | 12.44 | 11.03 | 11.74 | 2.23 |
| | Qwen2.5-Coder-7B-Instruct | 62.80 | 11.61 | 14.93 | 10.66 | 19.67 | 1.92 |
| | Llama-3.1-8B-Instruct | 11.86 | 51.82 | 30.02 | 6.30 | 1.45 | 1.53 |
| | Gemma-2-9b-it | 35.25 | 19.35 | 35.02 | 10.37 | 0.92 | 2.24 |
| | Phi-4 | 40.23 | 2.73 | 20.00 | 37.05 | 6.59 | 2.23 |
| | Phi-4-mini-instruct | 27.19 | 13.26 | 47.64 | 11.91 | 0.00 | 3.75 |
| | QwQ-32B | 30.37 | 0.83 | 13.64 | 55.17 | 8.68 | 1.68 |

formatting errors. For example, out of 30.63% cases where Qwen2.5-32B-Instruct for BDE does not generate a test-valid postcondition, 23.01% are due to compilation or formatting errors.

In Appendix A.2.3, we present ablation study on few-shot examples that demonstrates a strong positive correlation between the number of examples and postcondition quality. Additionally, we present ablation study on return types showing that postcondition generation effectiveness varies significantly across type categories, with numeric types achieving the highest validity (76.6%) and lowest compilation error rate (14.6%), while compound types (pointers, references, structs) present greater challenges with a 35.9% compilation error rate (see Appendix A.3 for detailed breakdown).

## 4.3 QUALITATIVE ANALYSIS

In this section, we introduce our methodology for assessing the correctness of specification synthesis in FUN2SPEC. Our approach combines automated oracles with systematic manual validation by three independent reviewers with formal methods expertise. Each specification was evaluated by two reviewers using standardized criteria for semantic correctness, completeness, and precision, with disagreements resolved by a third reviewer.

Classification of synthesized postconditions falls into the following categories:

- **Incorrect**: the test-valid candidate postcondition incorrectly captures the function intent.
- **Correct** but not strongest: the candidate postcondition correctly captures the function intent, but behavior is not fully specified.
- **Strongest**: the candidate postcondition was the strongest correct postcondition for the function.

We employ the following automated oracles to supplement manual assessment and classify LLM outputs:

- **Conditional behavior** must be specified using logical implication, so that program `if P then Q else R` is captured as $P \Rightarrow Q \vee \neg P \Rightarrow R$ in the post-condition.
- **Iterative loop behavior** is directly specified in the function post-condition using first order logic, so that a predicate P P can be specified as $\forall\, 0 \leq \texttt{i} < \texttt{sizeof(array)} : P(\texttt{array}[i])$ when P is true of all container elements, or $\exists\, 0 \leq \texttt{j} < \texttt{sizeof(array)} : \neg P(\texttt{array}[j])$ when at least one container element negates P P.

We distinguish between several types of correct candidates, whether they were trivially representing all possible executions, or they were unnecessarily too verbose and could be simplified. Through this rigorous evaluation process, FUN2SPEC generates remarkably accurate postconditions with only 2 incorrect cases, though some aren't the strongest. Listing 7 illustrates the distinction between correct and strongest postconditions. For function `containsDescriptor`, FUN2SPEC successfully generates the strongest postcondition by asserting that a `true` return implies the existence of a matching descriptor in the transition vector, while `false` implies no matches exist.

Figure 3: Classification of a sample of inferred postconditions

| Contract Type | Propo -sitional | First Order |
|---|---|---|
| Incorrect | 2 | 0 |
| Correct (Trivial) | 5 | 0 |
| Correct | 59 | 6 |
| Correct Strongest | 34 | 7 |
| Total | 100 | 13 |

```
1  // Generated Postcondition
2  (__out == true
3  ==> EXISTS(transitions.begin(), transitions.end(), it,
        descriptor == it->descriptor()))
4  && (__out == false
5  ==> FORALL(transitions.begin(), transitions.end(), it,
        descriptor != it->descriptor()))
6  static bool containsDescriptor(
7  const bsl::vector<baltzo::ZoneinfoTransition>&
        transitions,
8  const baltzo::LocalTimeDescriptor& descriptor
9  ) {
10     auto it  = transitions.begin();
11     auto end = transitions.end();
12     for (; it != end; ++it)
13         if (descriptor == it->descriptor())
14             return true;
15     return false;
16 }
```

Listing 7: Example of *Correct and Strongest* postcondition inferred with FUN2SPEC

## 5 RELATED WORK

**Classical Techniques:** Contract inference has received significant attention over two decades (Ernst, 2000; Ernst et al., 2007; Lahiri and Vanegue, 2011; Pandita et al., 2012; Nimmer and Ernst, 2002; Dillig et al., 2013). Static analysis approaches like Houdini (Lahiri and Vanegue, 2011; Nimmer and Ernst, 2002) infer pre and postconditions for C programs but require user-provided specification templates that are difficult to generalize (Dillig et al., 2013). Houdini's iterative check-and-refute cycle is scalable but presents challenges in understanding why specific candidates fail (Lahiri and Vanegue, 2011). Dynamic invariant detection tools like Daikon (Ernst et al., 2007; Ernst, 2000) overcome the template requirement by observing program executions. However, Daikon faces significant limitations with C++ codebases (Kusano et al., 2015). It struggles with complex memory management, pointer manipulation, and intricate data structures common in industrial C++ systems.

**LLM-based Approaches:** ML techniques were widely adopted to improve formal verification Garg et al. (2014); Si et al. (2020). More recently, LLMs have demonstrated remarkable capabilities for code generation and understanding (Chen et al., 2021; Xu et al., 2022; Ugare et al., 2024). Building on this foundation, researchers have leveraged LLMs for automated formal verification (Pei et al., 2023; Orenes-Vera et al., 2023; First et al., 2023; Ma et al., 2024; Wen et al., 2024; He et al., 2024; Wu et al., 2024; Lahiri, 2024; Ma et al., 2024; Ruan et al., 2024; Liu et al., 2025; Yang et al., 2025). Significant progress has emerged in generating formal contracts using LLMs, including preconditions (Dinella et al., 2024), postconditions (Endres et al., 2024a), and invariants (Pei et al., 2023; Pirzada et al., 2024; Sun et al., 2025) and more specifically inductive loop invariants (Kamath et al., 2023; Yu et al., 2023; Liu et al., 2024b;a). The most relevant prior work, NL2POST (Endres et al., 2024a), focuses on inferring postconditions from function implementations and natural language comments, but it is limited to small, standalone Python functions from HumanEval (Chen et al., 2021). Their method achieves only 20–30% test-validity with open-source models. In contrast, FUN2SPEC targets more realistic C++ codebases and consistently outperforms NL2POST across both HumanEval-CPP and FormalSpecCPP benchmarks. Specifically, FUN2SPEC achieves average improvements of 19.2 and 20.1 percentage points over NL2POST on HumanEval-CPP and FormalSpecCPP, respectively, with test-validity reaching up to 76% on the strongest model.

**Limitations** While verification competitions like SV-COMP (Beyer, 2024) and frameworks such as CBMC (Kroening et al., 2023), SMACK (Carter et al., 2016), and SeaHorn (Gurfinkel et al., 2015) have made significant progress in program verification, these tools still struggle with large modern C++ codebases due to complex language features and scale limitations. There have been efforts to incorporate formal specifications directly into the C++ standard, with proposals for contract programming features (Doumler and Krzemieński, 2025). However, these standardization efforts remain ongoing and not yet widely implemented. Consequently, FUN2SPEC generated specifications cannot be verified automatically and we rely on test-validity as a proxy for formal verification.

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

# A APPENDIX.

## A.1 PROMPT INSTRUCTIONS

```
1  As an expert language model trained in understanding code, your task is to generate a
       postcondition (POST) for the provided C++ function.
2  A postcondition is a predicate wrapped in POST(any_predicate) that represents a
       condition guaranteed to be true after the function returns.
3  Follow these rules to construct the postcondition:
4
5  1. Syntax: Wrap the postcondition in POST(any_predicate).
6  2. Implications: Use the symbol "==>" for logical implication. For example, condition1
       ==> condition2 indicates that if condition1 is true, then condition2 must also be
       true.
7  3. Logical Operators: You may use "&&" (logical AND) and "||" (logical OR) to combine
       multiple conditions within a single predicate. condition1 && condition2 indicates
       that both conditions must be true. condition1 || condition2 indicates that at least
       one of the conditions is true.
8  4. Quantifiers: You may use EXISTS and FORALL to express quantified conditions:
9     - EXISTS(start, end, var, condition): There exists a value var in range [start, end)
          that satisfies condition
10    - FORALL(start, end, var, condition): All values var in range [start, end) satisfy
          condition
11
12    Example: POST(res_tmp == true ==> EXISTS(0, numbers.size(), i, numbers[i] == target))
13 5. All predicates used in postcondition should be valid C++ expressions. All predicates
       will be executed using C++ compiler.
14 6. Valid Function Names: All function/method calls used in the predicate should exist in
       the context of the function. Do not hallucinate use and any hypothetical function
       name! Instead give a simpler postcondition.
15 7. Naming the Return Value: Use "res_tmp" as the name of the return value.
16 8. Trivial Postcondition: If no specific predicates must hold for the function, return a
       trivial postcondition, POST(true).
17 9. Single Postcondition: Return only one postcondition per function.
18 10. Always include the appropriate namespace in postconditions if the constant, type, or
        function is qualified with a namespace in the code. If no namespace is used in the
        code, refer to the constant or type directly without a namespace in the
        postcondition.
```

Listing 8: Postcondition Generation Instruction

## A.2 ABLATION STUDY

### A.2.1 POSTCONDITION COMPLEXITY.

Figure 4 illustrates the distribution of the number of atomic expressions present in the generated postconditions for the model Qwen/Qwen2.5-32B-Instruct on repository BDE. The x-axis represents the number of atoms in each predicate, while the y-axis indicates the count of postconditions containing the corresponding number of atoms. We observe a wide range of complexity in the generated postconditions, as the atoms in the generated postconditions range from 1 to 16.

### A.2.2 FEEDBACK AND QUANTIFIERS.

Table 4 presents the effectiveness of FUN2SPEC in generating valid postconditions across both the HumanEval-CPP and FormalSpecCPP benchmarks. We evaluate each model under three settings: (1)

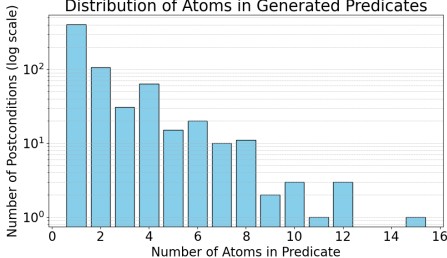

Figure 4: Distribution of the number of atoms in the generated predicates for the model Qwen2.5-32B-Instruct. The y-axis is logarithmic, showing the frequency of postconditions with varying atom counts.

Table 4: Postcondition test-valid percentage for each model on HumanEval and FormalSpecCPP benchmarks across three settings.

| Model | Benchmark | Postcondition test-valid (%) | | |
|---|---|---|---|---|
| | | Standard Setting | Reprompt Off | Quantifiers + Reprompt Off |
| Qwen3-32B | HumanEval | **55.83** | 52.76 | 52.15 |
| | FormalSpecCPP | **74.00** | 74.51 | 57.84 |
| Qwen2.5-32B | HumanEval | **63.19** | 47.85 | 48.47 |
| | FormalSpecCPP | **76.00** | 74.51 | 50.98 |
| Qwen2.5-Coder-7B | HumanEval | **52.15** | 39.88 | 46.63 |
| | FormalSpecCPP | **67.65** | 56.86 | 54.90 |
| Llama-3.1-8B | HumanEval | **17.18** | 0.61 | 1.23 |
| | FormalSpecCPP | **20.00** | 0.98 | 0.98 |
| Gemma-2-9b-it | HumanEval | **36.20** | 33.13 | 28.22 |
| | FormalSpecCPP | 59.00 | **60.78** | 40.20 |
| Phi-4-mini | HumanEval | **25.15** | 22.70 | 20.86 |
| | FormalSpecCPP | **43.14** | 38.24 | 33.33 |
| Phi-4 | HumanEval | 22.09 | **22.70** | 19.63 |
| | FormalSpecCPP | **57.84** | 53.92 | 38.24 |
| QwQ-32B | HumanEval | 6.13 | **12.88** | 12.27 |
| | FormalSpecCPP | **48.04** | 43.14 | 19.61 |

Table 5: FUN2SPEC performance with varying number of few-shot examples with Qwen2.5-32B-Instruct

| Few-shot Examples | Test Valid (%) | Test Invalid (%) | Compilation Error (%) |
|---|---|---|---|
| 4 | 69.49 | 6.36 | 23.33 |
| 3 | 69.22 | 8.31 | 22.15 |
| 2 | 60.91 | 9.93 | 27.85 |
| 1 | 57.42 | 9.62 | 30.18 |
| 0 | 43.06 | 8.29 | 32.70 |

our standard FUN2SPEC setting with all features enabled, (2) with reprompting disabled (error feedback removed), and (3) with both quantifiers and reprompting disabled. The reprompting mechanism proves crucial for most models. Additionally, quantifier support significantly impacts effectiveness, where disabling quantifiers causes performance decreases of up to 25 percentage points, highlighting the importance of supporting rich specification language features when inferring postconditions.

### A.2.3 FEW-SHOT EXAMPLES.

Table 5 presents the impact of the number of few-shot examples on performance with Qwen2.5-32B-Instruct. As the number of few-shot examples increases, the percentage of test-valid results consistently improves. For instance, with 4 few-shot examples, the test-valid rate is the highest at 69.49%, whereas with 0 examples, it drops significantly to 43.06%. This trend indicates that providing more examples significantly enhances the FUN2SPEC's ability to produce valid postconditions.

### A.3 EFFECT OF RETURN TYPES ON GENERATION

We aggregate testing results for the Qwen2.5-32B-Instruct model into three main categories based on the return type of the function: Numeric Types, Compound Types (pointers, references, structs), and Other Types (booleans, enums, char). As shown in Table 6, FUN2SPEC achieves a 76.6% success rate for numeric types, with a relatively low compilation error rate of 14.6%. Generating postconditions for compound types is challenging, as FUN2SPEC encounters a 35.9% compilation error rate due to the complexity of pointers and references, though it still maintains a 60.8% validity. Other Types, including miscellaneous categories such as enums and character types, showed moderate performance with a 65.5% validity, 29.7% compilation errors, and a 4.7% failure rate.

Table 6: Postcondition validation results categorized by the return type of functions. The counts represent the number of valid, invalid, and failed-to-compile (Compilation Error) postconditions.

| Category | Test Valid (%) | Test Invalid (%) | Compilation Error (%) |
|---|---|---|---|
| Numerical Types | 76.6 | 14.6 | 8.8 |
| Compound Types | 60.8 | 35.9 | 3.3 |
| Other Types | 65.5 | 29.7 | 4.7 |

## A.4 SCALING THE REFINEMENT ITERATIONS ON BENCHMARKS

Table 7: Test-Validity (%) comparison between FUN2SPEC (across retries) and NL2POST on HumanEval-CPP

| Model | NL2POST | FUN2SPEC 1 retry | FUN2SPEC 5 retries | FUN2SPEC 10 retries ($\Delta$) |
|---|---|---|---|---|
| Qwen2.5-32B | 20.2 | 63.2 | 74.8 | 76.1 (+55.9) |
| Qwen2.5-Coder-7B | 30.7 | 52.1 | 65.3 | 67.5 (+36.8) |

Table 8: Test-Validity (%) comparison between FUN2SPEC (across retries) and NL2POST on Formal-SpecCPP

| Model | NL2POST | FUN2SPEC 1 retry | FUN2SPEC 5 retries | FUN2SPEC 10 retries ($\Delta$) |
|---|---|---|---|---|
| Qwen2.5-32B | 43.1 | 76.0 | 80.5 | 80.5 (+37.4) |
| Qwen2.5-Coder-7B | 52.9 | 67.6 | 69.6 | 70.6 (+17.7) |

As shown in Tables 7 and 8, on both HumanEval-CPP and FormalSpecCPP, FUN2SPEC consistently outperforms the NL2POST baseline by a large margin. Even with just 1 retry, FUN2SPEC already surpasses NL2POST by 20–40 percentage points. Increasing retries further improves performance: gains plateau by 5–10 retries, but the improvements remain substantial. For example, on HumanEval-CPP, Qwen2.5-32B improves from 20.2% (NL2POST) to 76.1% (FUN2SPEC, 10 retries), a +55.9 point gain.

## A.5 SCALING THE REFINEMENT ITERATIONS ON LARGE CODEBASES

As shown in Table 9, scaling retries with FUN2SPEC substantially improves test validity while runtime grows sublinearly with the number of retries. For Qwen2.5-32B-Instruct, validity rises from 69.4% (1 retry) to 86.9% (10 retries), with compilation and formatting errors reduced by over half. Similarly, Qwen2.5-Coder-7B-Instruct improves from 57.6% to 73.5%. These results show that retries yield strong improvements in accuracy with relatively moderate additional computational cost.

## A.6 QUALITATIVE ANALYSIS

In this section, we classify 100 generated postconditions on BDE with `Qwen2.5-32B-Instruct` model.

## A.7 POSTCONDITIONS WITH QUANTIFIERS

1. **Filename:** baljsn_datumutil.cpp, **Function:** int encodeArray, **Classification:** correct

```
int encodeArraycorrect
(__out == 0 ==  FORALL(0, datum.length(), i, u::encodeValue(formatter, datum[i],
strictTypesCheckStatus) == 0))
&& (__out != 0 ==  EXISTS(0, datum.length(), i, u::encodeValue(formatter, da-
tum[i], strictTypesCheckStatus) != 0))
```

2. **Filename:** baljsn_datumutil.cpp, **Function:** int encodeObject, **Classification:** correct

Table 9: Model Performance with FUN2SPEC after scaling the number of retries on BDE

| Model Name | Test Valid (%) | Test Invalid (%) | Compilation Error (%) | Formatting Error (%) | Trivial (%) | Avg. Atoms | Time (s) |
|---|---|---|---|---|---|---|---|
| Qwen2.5-32B-Instruct (1 retry) | 69.41 | 6.09 | 14.80 | 9.70 | 14.97 | 2.29 | 628.3 |
| Qwen2.5-32B-Instruct (5 retries) | 84.52 | 4.81 | 6.98 | 3.69 | 6.37 | 3.21 | 1247.2 |
| Qwen2.5-32B-Instruct (10 retries) | 86.94 | 4.81 | 4.56 | 3.69 | 5.78 | 3.22 | 1831.9 |
| Qwen2.5-Coder-7B-Instruct (1 retry) | 57.61 | 19.48 | 12.77 | 10.15 | 12.93 | 1.86 | 773.42 |
| Qwen2.5-Coder-7B-Instruct (5 retries) | 72.16 | 19.48 | 12.77 | 10.15 | 6.78 | 2.36 | 1585.8 |
| Qwen2.5-Coder-7B-Instruct (10 retries) | 73.45 | 19.48 | 12.77 | 10.15 | 6.42 | 2.41 | 2485.3 |

> int encodeObjectcorrect
> (__out != 0 == EXISTS(0, datum.size(), i, u::encodeValue(formatter, datum[i].value(), strictTypesCheckStatus, &datum[i].key()) != 0))
> && (FORALL(0, datum.size(), i, u::encodeValue(formatter, datum[i].value(), strictTypesCheckStatus, &datum[i].key()) == 0) == __out == 0)

3. **Filename:** ball_managedattributeset.cpp, **Function:** bool ManagedAttributeSet::evaluate, **Classification:** correct strongest

> bool ManagedAttributeSet::evaluatecorrect strongest
> __out == true == FORALL(begin(), end(), iter, containerList.hasValue(iter-attribute()))

4. **Filename:** ball_managedattributeset.cpp, **Function:** bool ball::operator==, **Classification:** correct strongest

> bool ball::operator==correct strongest
> __out == true == (lhs.numAttributes() == rhs.numAttributes() && FORALL(lhs.begin(), lhs.end(), attr, rhs.isMember(attr)))

5. **Filename:** ball_recordjsonformatter.cpp, **Function:** int FixedFieldFormatter::parse, **Classification:** correct

> int FixedFieldFormatter::parsecorrect
> (__out == 0 == FORALL(0, v.size(), i, v[i].value().isString())) && (__out == -1 == EXISTS(0, v.size(), i, !v[i].value().isString()))

6. **Filename:** ball_ruleset.cpp, **Function:** bool ball::operator==, **Classification:** correct strongest

> bool ball::operator==correct strongest
> (__out == true == FORALL(0, lhs.numRules(), i, rhs.ruleId(*lhs.getRuleById(i))>= 0))
> && (__out == false == lhs.numRules() != rhs.numRules() || EXISTS(0, lhs.numRules(), i, !(rhs.ruleId(*lhs.getRuleById(i))>= 0)))

7. **Filename:** baltzo_zoneinfo.cpp, **Function:** static bool containsDescriptor, **Classification:** correct strongest

> static bool containsDescriptorcorrect strongest
> (__out == true == EXISTS(transitions.begin(), transitions.end(), it, descriptor ==
> it- descriptor()))
> && (__out == false == FORALL(transitions.begin(), transitions.end(), it, descriptor
> != it- descriptor()))

8. **Filename:** baltzo_zoneinfobinaryreader.cpp, **Function:** static bool areAllPrintable, **Classification:** correct strongest

> static bool areAllPrintablecorrect strongest
> (__out == true == FORALL(0, length, i, bdlb::CharType::isPrint(buffer[i])))
> && (__out == false == EXISTS(0, length, i, !bdlb::CharType::isPrint(buffer[i])))

9. **Filename:** balxml_prefixstack.cpp, **Function:** const PredefinedPrefix& lookupPredefinedPrefix, **Classification:** correct strongest

> const PredefinedPrefix& lookupPredefinedPrefixcorrect strongest
> FORALL(0, ARRAY_LEN(predefinedPrefixes), i, prefix == predefinedPrefixes[i].d_prefix == &__out == &predefinedPrefixes[i])
> || &__out == &nullPrefix

10. **Filename:** bdlc_indexclerk.cpp, **Function:** areInvariantsPreserved, **Classification:** correct

> areInvariantsPreservedcorrect
> __out == true == FORALL(0, unusedStack.size(), i, 0 ¡= unusedStack[i] && unusedStack[i] ¡ nextNewIndex && bin[unusedStack[i]] ¡ 2)

11. **Filename:** bldc_charconvertutf16.cpp, **Function:** const OctetType *skipContinuations, **Classification:** vacuous

> const OctetType *skipContinuationsvacuous
> FORALL(octets, __out, i, (*i & CONTINUE_MASK) != CONTINUE_Classification)

12. **Filename:** bdlt_fixutil.cpp, **Function:** int asciiToInt, **Classification:** correct

> int asciiToIntcorrect
> (__out == 0 == (*nextPos == end && *result == tmp)) && (__out == -1 == !FORALL(begin, end, i, isdigit(*i)))

13. **Filename:** bdlt_timetable.cpp, **Function:** Timetable::const_iterator Timetable::begin(), **Classification:** correct strongest

> Timetable::const_iterator Timetable::begin()correct strongest
> FORALL(0, __out.dayIndex(), i, d_timetable[i].size() == 0)

## A.8 POSTCONDITIONS WITHOUT QUANTIFIERS

1. **Filename:** balb_controlmanager.cpp, **Function:** ControlManager::registerHandler, **Classification:** correct

> ControlManager::registerHandlercorrect
> __out == 0 || __out == 1

2. **Filename:** balb_leakybucket.cpp, **Function:** calculateNumberOfUnitsToDrain, **Classification:** correct

> calculateNumberOfUnitsToDraincorrect
> __out >= 0 && (*fractionalUnitDrainedInNanoUnits ¡ k_NANOUNITS_PER_UNIT)

3. **Filename:** balb_leakybucket.cpp, **Function:** calculateTimeToSubmit, **Classification:** correct

> calculateTimeToSubmitcorrect
> __out >= bsls::TimeInterval(0, 0)

4. **Filename:** balb_performancemonitor.cpp, **Function:** nearlyEqual, **Classification:** correct strongest

> nearlyEqualcorrect strongest
> __out == (bsl::fabs(lhs - rhs) ¡ bsl::numeric_limits¡double¿::epsilon())

5. **Filename:** balb_pipetaskmanager.cpp, **Function:** makeControlChannel, **Classification:** correct

> makeControlChannelcorrect
> __out != NULL

6. **Filename:** balb_ratelimiter.cpp, **Function:** RateLimiter::calculateTimeToSubmit, **Classification:** correct

> RateLimiter::calculateTimeToSubmitcorrect
> __out >= timeToSubmitPeak && __out >= timeToSubmitSustained

7. **Filename:** balber_berdecoder.cpp, **Function:** BerDecoder_Node::startPos, **Classification:** correct strongest

> BerDecoder_Node::startPoscorrect strongest
> __out >= 0

8. **Filename:** balber_berdecoderoptions.cpp, **Function:** BerDecoderOptions::lookupAttributeInfo, **Classification:** correct strongest

> BerDecoderOptions::lookupAttributeInfo
> [big string]

9. **Filename:** balber_berdecoder.cpp, **Function:** BerEncoder::logError, **Classification:** correct strongest

> BerEncoder::logErrorcorrect strongest
> static_cast¡int¿(__out) >= static_cast¡int¿(BloombergLP::balber::BerEncoder::e_BER_ERROR)

10. **Filename:** balber_beruniversalClassificationnumber.cpp, **Function:** BerUniversalClassificationNumber::toString, **Classification:** correct

> BerUniversalClassificationNumber::toStringcorrect
> __out != NULL

11. **Filename:** balber_berutil.cpp, **Function:** ReadRestFunctor::operator(), **Classification:** correct

> ReadRestFunctor::operator()correct
> __out>= d_oldSize && __out ¡= newSize

12. **Filename:** balber_berutil.cpp, **Function:** BerUtil_IdentifierImpUtil::getIdentifierOctets, **Classification:** correct

> BerUtil_IdentifierImpUtil::getIdentifierOctetscorrect
> __out == SUCCESS || __out == FAILURE

13. **Filename:** balber_berutil.cpp, **Function:** BerUtil_IdentifierImpUtil::putIdentifierOctets, **Classification:** correct

> BerUtil_IdentifierImpUtil::putIdentifierOctetscorrect
> __out == SUCCESS || __out == FAILURE

14. **Filename:** balber_berutil.cpp, **Function:** BerUtil_IntegerImpUtil::getNumOctetsToStream, **Classification:** correct

> BerUtil_IntegerImpUtil::getNumOctetsToStreamcorrect
> (value == 0 == __out == 1) && (value != 0 == __out> 0)

15. **Filename:** balber_berutil.cpp, **Function:** BerUtil_TimezoneOffsetImpUtil::isValidTimezoneOffsetInMinutes, **Classification:** correct strongest

> BerUtil_TimezoneOffsetImpUtil::isValidTimezoneOffsetInMinutescorrect strongest
> (__out == true == (k_MIN_OFFSET ¡= value && value ¡= k_MAX_OFFSET)) &&
> (__out == false == (value ¡ k_MIN_OFFSET || value> k_MAX_OFFSET))

16. **Filename:** balcl_commandline.cpp, **Function:** EnvironmentVariableAccessor::value(), **Classification:** correct strongest

> EnvironmentVariableAccessor::value()correct strongest
> __out == d_returnValue

17. **Filename:** balcl_commandline.cpp, **Function:** bsl::ostream& u::operator¡¡, **Classification:** correct

> bsl::ostream& u::operator¡¡correct
> & __out == &stream

18. **Filename:** balcl_commandline.cpp, **Function:** isValidEnvironmentVariableName, **Classification:** correct but overfit

> isValidEnvironmentVariableNamecorrect but overfit
> [big string]

19. **Filename:** balcl_commandline.cpp, **Function:** parseEnvironmentVariable, **Classification:** correct

> parseEnvironmentVariablecorrect
> (__out == -1) || (__out == 1) || (__out>= 0)

20. **Filename:** balcl_commandline.cpp, **Function:** CommandLine::operator=, **Classification:** correct

> CommandLine::operator=correct
> &__out == this

21. **Filename:** balcl_commandline.cpp, **Function:** CommandLine::hasOption, **Classification:** correct strongest

> CommandLine::hasOptioncorrect strongest
> (findName(name)>= 0 == __out == true) && (findName(name) ¡ 0 == __out == false)

22. **Filename:** balcl_commandline.cpp, **Function:** balcl::operator==, **Classification:** correct strongest

> balcl::operator==correct strongest
> __out == (lhs.isParsed() && rhs.isParsed() && lhs.options() == rhs.options())

23. **Filename:** balcl_commandline.cpp, **Function:** CommandLineOptionsHandle::index, **Classification:** correct

> CommandLineOptionsHandle::indexcorrect
> __out>= -1

24. **Filename:** balcl_occurrenceinfo.cpp, **Function:** OccurrenceInfo::operator=, **Classification:** correct strongest

> OccurrenceInfo::operator=correct strongest
> &__out == this && d_defaultValue == rhs.d_defaultValue &&
> d_isRequired == rhs.d_isRequired && d_isHidden == rhs.d_isHidden

25. **Filename:** balcl_occurrenceinfo.cpp, **Function:** balcl::operator==, **Classification:** correct strongest

> balcl::operator==correct strongest
> [big string]

26. **Filename:** balcl_option.cpp, **Function:** Option::operator=, **Classification:** correct

> Option::operator=correct
> &__out != nullptr

27. **Filename:** balcl_option.cpp, **Function:** balcl::operator==, **Classification:** trivial

> balcl::operator==trivial
> __out == true || __out == false

28. **Filename:** balcl_optioninfo.cpp, **Function:** bsl::ostream& balcl::operator¡¡, **Classification:** correct

> bsl::ostream& balcl::operator¡¡correct
> &__out == &stream

29. **Filename:** balcl_optiontype.cpp, **Function:** bsl::ostream& OptionType::print, **Classification:** correct

> bsl::ostream& OptionType::printcorrect
> &__out == &stream

30. **Filename:** balcl_typeinfo.cpp, **Function:** const char *elemTypeToString, **Classification:** correct

> const char *elemTypeToStringcorrect
> __out != NULL

31. **Filename:** balcl_typeinfo.cpp, **Function:** OptionType::Enum_BoolConstraint::type(), **Classification:** correct strongest

> OptionType::Enum_BoolConstraint::type()correct strongest
> __out == OptionType::e_BOOL

32. **Filename:** balcl_typeinfo.cpp, **Function:** TypeInfo& TypeInfo::operator=, **Classification:** correct

> TypeInfo& TypeInfo::operator=correct
> &__out == this

33. **Filename:** balcl_typeinfo.cpp, **Function:** bool balcl::operator==, **Classification:** correct strongest

> bool balcl::operator==correct strongest
> [big string]

34. **Filename:** baljsn_datumutil.cpp, **Function:** int decodeObject, **Classification:** correct

> int decodeObjectcorrect
> __out == 0 || __out == -1 || __out == -2 || __out == -3 || __out == -4

35. **Filename:** baljsn_datumutil.cpp, **Function:** int decodeArray, **Classification:** correct

```
int decodeArraycorrect
(maxNestedDepth ¡ 0 == __out == -4)
&& (tokenizer-¿tokenType() == baljsn::Tokenizer::e_ERROR == __out == -1)
&& (decodeValue(&elementValue, errorStream, tokenizer, maxNestedDepth) != 0
== __out == -2)
&& (__out == 0 || __out == -4 || __out == -1 || __out == -2)
```

36. **Filename:** baljsn_datumutil.cpp, **Function:** int extractValue, **Classification:** correct

```
int extractValuecorrect
__out == 0 || __out == -1
```

37. **Filename:** baljsn_datumutil.cpp, **Function:** int DatumUtil::decode, **Classification:** correct

```
int DatumUtil::decodecorrect
__out == 0 || __out == -1 || __out == -2 || __out == -3
```

38. **Filename:** baljsn_decoder.cpp, **Function:** bsl::ostream& Decoder::logTokenizerError, **Classification:** correct

```
bsl::ostream& Decoder::logTokenizerErrorcorrect
&__out == &d_logStream
```

39. **Filename:** baljsn_encoder.cpp, **Function:** int Encoder_EncodeImplUtil::encodeCharArray, **Classification:** trivial

```
int Encoder_EncodeImplUtil::encodeCharArraycorrect trivial
__out>= 0 || __out < 0
```

40. **Filename:** ball_administration.cpp, **Function:** int Administration::addCategory, **Classification:** trivial

```
int Administration::addCategorycorrect trivial
__out == 0 || __out == 1
```

41. **Filename:** ball_asyncfileobserver.cpp, **Function:** bool isStopRecord, **Classification:** correct strongest

```
bool isStopRecordcorrect strongest
__out == (0 == record.d_record.get())
```

42. **Filename:** ball_attribute.cpp, **Function:** int Attribute::hash, **Classification:** correct

```
int Attribute::hashcorrect
0 ¡= __out && __out < size
```

43. **Filename:** ball_attribute.cpp, **Function:** bsl::ostream& Attribute::print, **Classification:** correct

```
bsl::ostream& Attribute::printcorrect
&__out == &stream
```

44. **Filename:** ball_attributecollectorregistry.cpp, **Function:** int AttributeCollectorRegistry::addCollector, **Classification:** correct

> int AttributeCollectorRegistry::addCollectorcorrect
> __out == 0 || __out == 1

45. **Filename:** ball_attributecontainerlist.cpp, **Function:** AttributeContainerList& AttributeContainerList::operator=, **Classification:** correct

> AttributeContainerList& AttributeContainerList::operator=correct
> &__out == this

46. **Filename:** ball_attributecontainerlist.cpp, **Function:** bool ball::operator==, **Classification:** correct strongest

> bool ball::operator==correct strongest
> __out == (lhs.numContainers() == rhs.numContainers() &&
> std::equal(lhs.begin(), lhs.end(), rhs.begin()))

47. **Filename:** ball_attributecontainerlist.cpp, **Function:** RuleSet::MaskType AttributeContext_RuleEvaluationCache::update, **Classification:** correct

> RuleSet::MaskType AttributeContext_RuleEvaluationCache::updatecorrect
> __out>= 0

48. **Filename:** ball_attributecontainerlist.cpp, **Function:** bsl::ostream& AttributeContext_RuleEvaluationCache::print, **Classification:** correct

> bsl::ostream& AttributeContext_RuleEvaluationCache::printcorrect
> &__out == &stream

49. **Filename:** ball_attributecontainerlist.cpp, **Function:** const bslmt::ThreadUtil::Key& AttributeContext::contextKey, **Classification:** correct

> const bslmt::ThreadUtil::Key& AttributeContext::contextKeycorrect
> &__out == &s_contextKey

50. **Filename:** ball_attributecontainerlist.cpp, **Function:** AttributeContext *AttributeContext::getContext, **Classification:** correct

> AttributeContext *AttributeContext::getContextcorrect
> __out != NULL

51. **Filename:** ball_attributecontainerlist.cpp, **Function:** bsl::ostream& AttributeContext::print, **Classification:** correct

> bsl::ostream& AttributeContext::printcorrect
> &__out == &stream

52. **Filename:** ball_broadcastobserver.cpp, **Function:** int BroadcastObserver::deregisterObserver, **Classification:** correct

> int BroadcastObserver::deregisterObservercorrect
> __out == 0 || __out == 1

53. **Filename:** ball_broadcastobserver.cpp, **Function:** bsl::shared_ptr¡const Observer¿ BroadcastObserver::findObserver, **Classification:** correct strongest

> bsl::shared_ptr¡const Observer¿ BroadcastObserver::findObservercorrect strongest
> (__out.use_count()> 0) || (__out.get() == nullptr)

54. **Filename:** ball_category.cpp, **Function:** int Category::setLevels, **Classification:** correct

> int Category::setLevelscorrect
> __out == 0 || __out == -1

55. **Filename:** ball_categorymanager.cpp, **Function:** Category *CategoryManager::addNewCategory, **Classification:** correct

> Category *CategoryManager::addNewCategorycorrect
> __out != NULL

56. **Filename:** ball_categorymanager.cpp, **Function:** const Category *CategoryManager::lookupCategory, **Classification:** correct strongest

> const Category *CategoryManager::lookupCategorycorrect strongest
> (__out != 0 == d_registry.find(categoryName) != d_registry.end()) &&
> (__out == 0 == d_registry.find(categoryName) == d_registry.end())

57. **Filename:** ball_context.cpp, **Function:** bool Context::isValid, **Classification:** incorrect

> bool Context::isValidincorrect
> !(transmissionCause == Transmission::e_PASSTHROUGH
> && __out == (recordIndex == 0 && sequenceLength == 1))

58. **Filename:** ball_context.cpp, **Function:** bsl::ostream& Context::print, **Classification:** correct

> bsl::ostream& Context::printcorrect
> &__out == &stream

59. **Filename:** ball_defaultattributecontainer.cpp, **Function:** DefaultAttributeContainer& DefaultAttributeContainer::operator=, **Classification:** correct strongest

> DefaultAttributeContainer& DefaultAttributeContainer::operator=correct strongest
> &__out == this && &__out == &rhs

60. **Filename:** ball_defaultattributecontainer.cpp, **Function:** bool ball::operator==, **Classification:** incorrect

> bool ball::operator==incorrect
> !(lhs.numAttributes() != rhs.numAttributes() == !__out)
> && (lhs.numAttributes() == rhs.numAttributes()
> && std::all_of(lhs.begin(), lhs.end(), [&rhs](const auto& attr) return
> rhs.hasValue(attr); ) == __out)

61. **Filename:** ball_fileobserver.cpp, **Function:** bslma::Allocator *FileObserver::allocator, **Classification:** correct strongest

> bslma::Allocator *FileObserver::allocatorcorrect strongest
> __out != NULL

62. **Filename:** ball_fileobserver2.cpp, **Function:** static int getErrorCode, **Classification:** correct strongest

> static int getErrorCodecorrect strongest
> __out >= 0

63. **Filename:** ball_fileobserver2.cpp, **Function:** static int openLogFile, **Classification:** correct

> static int openLogFilecorrect
> __out == 0 || __out == -1

64. **Filename:** ball_fileobserver2.cpp, **Function:** static bdlt::Datetime computeNextRotationTime, **Classification:** correct

> static bdlt::Datetime computeNextRotationTimecorrect
> __out >= fileCreationTimeUtc && (fuzzyEqual(referenceStartTime, fileCreationTimeUtc, interval)
> == __out >= fileCreationTimeUtc + interval)

65. **Filename:** ball_log.cpp, **Function:** Log::format, **Classification:** correct strongest

> Log::formatcorrect strongest
> ((bsl::size_t)__out >= numBytes == __out == -1)
> && ((bsl::size_t)__out < numBytes == __out != -1)

66. **Filename:** ball_loggercategoryutil.cpp, **Function:** Category *LoggerCategoryUtil::addCategoryHierarchically, **Classification:** correct strongest

> Category *LoggerCategoryUtil::addCategoryHierarchicallycorrect strongest
> (__out == 0) || (__out != 0 && loggerManager-¿lookupCategory(categoryName) == __out)

67. **Filename:** ball_loggermanager.cpp, **Function:** Record *RecordSharedPtrUtil::disassembleSharedPtr, **Classification:** correct

> Record *RecordSharedPtrUtil::disassembleSharedPtrcorrect
> __out != nullptr

68. **Filename:** ball_loggermanager.cpp, **Function:** const char *filterName, **Classification:** correct strongest

> const char *filterNamecorrect strongest
> (nameFilter ? __out == filteredNameBuffer-¿c_str() : __out == originalName)

69. **Filename:** ball_loggermanager.cpp, **Function:** inline static ball::Severity::Level convertBslsLogSeverity, **Classification:** correct

> inline static ball::Severity::Level convertBslsLogSeveritycorrect
> (severity == bsls::LogSeverity::e_FATAL == __out == ball::Severity::e_FATAL)

70. **Filename:** ball_loggermanager.cpp, **Function:** bsl::shared_ptr¡Record¿ Logger::getRecordPtr, **Classification:** correct strongest

> bsl::shared_ptr¡Record¿ Logger::getRecordPtrcorrect strongest
> __out-¿fixedFields().getFileName() == fileName &&
> __out-¿fixedFields().getLineNumber() == lineNumber

71. **Filename:** ball_loggermanager.cpp, **Function:** Record *Logger::getRecord, **Classification:** trivial

> Record *Logger::getRecordcorrect trivial
> __out != nullptr

72. **Filename:** ball_loggermanager.cpp, **Function:** bool LoggerManager::isCategoryEnabled, **Classification:** correct strongest

> bool LoggerManager::isCategoryEnabledcorrect strongest
> (category-¿relevantRuleMask() && __out == (ThresholdAggregate::maxLevel(levels)¿= severity))
> || (!category-¿relevantRuleMask() && __out == (category-¿maxLevel()¿= severity))

73. **Filename:** ball_loggermanagerconfiguration.cpp, **Function:** LoggerManagerConfiguration::operator=, **Classification:** correct strongest

> LoggerManagerConfiguration::operator=correct strongest
> __out.d_defaults == rhs.d_defaults && __out.d_userPopulator == rhs.d_userPopulator
> && __out.d_categoryNameFilter == rhs.d_categoryNameFilter
> && __out.d_defaultThresholdsCb == rhs.d_defaultThresholdsCb
> && __out.d_logOrder == rhs.d_logOrder && __out.d_triggerMarkers == rhs.d_triggerMarkers

74. **Filename:** ball_loggermanagerconfiguration.cpp, **Function:** const LoggerManagerDefaults& LoggerManagerConfiguration::defaults(), **Classification:** correct strongest

> const LoggerManagerDefaults& LoggerManagerConfiguration::defaults()correct
> strongest
> &__out == &d_defaults

75. **Filename:** ball_loggermanagerdefaults.cpp, **Function:** bool LoggerManagerDefaults::isValidDefaultRecordBufferSize, **Classification:** correct strongest

> bool LoggerManagerDefaults::isValidDefaultRecordBufferSize correct strongest
> __out == (0 < numBytes)

76. **Filename:** ball_managedattribute.cpp, **Function:** bsl::ostream& ManagedAttribute::print, **Classification:** correct

> bsl::ostream& ManagedAttribute::printcorrect
> &__out == &stream

77. **Filename:** ball_managedattributeset.cpp, **Function:** int ManagedAttributeSet::hash, **Classification:** correct strongest

> int ManagedAttributeSet::hashcorrect strongest
> (0 <= __out) && (__out < size)

78. **Filename:** ball_managedattributeset.cpp, **Function:** ManagedAttributeSet& ManagedAttributeSet::operator=, **Classification:** correct

> ManagedAttributeSet& ManagedAttributeSet::operator=correct
> this->d_attributeSet == rhs.d_attributeSet

79. **Filename:** ball_managedattributeset.cpp, **Function:** bool ManagedAttributeSet::evaluate, **Classification:** correct strongest

> bool ManagedAttributeSet::evaluatecorrect strongest
> (__out == true ==
> std::all_of(begin(), end(), [&](auto& attr) return containerList.hasValue(attr.attribute()); ))
> && (__out == false == std::any_of(begin(), end(), [&](auto& attr)
> return !containerList.hasValue(attr.attribute()); ))

80. **Filename:** ball_record.cpp, **Function:** bsl::ostream& Record::print, **Classification:** correct

> bsl::ostream& Record::printcorrect
> &__out == &stream && __out->rdstate() == std::ios_base::goodbit

81. **Filename:** ball_recordattributes.cpp, **Function:** ball_recordattributes.cpp, **Classification:** correct strongest

> ball_recordattributes.cppcorrect strongest
> (lhs.d_timestamp == rhs.d_timestamp && lhs.d_processID == rhs.d_processID
> && lhs.d_threadID == rhs.d_threadID && lhs.d_severity == rhs.d_severity &&
> lhs.d_lineNumber == rhs.d_lineNumber && lhs.d_fileName == rhs.d_fileName &&
> lhs.d_category == rhs.d_category && lhs.messageRef() == rhs.messageRef()) ==
> __out

82. **Filename:** ball_recordjsonformatter.cpp, **Function:** int FixedFieldFormatter::parse, **Classification:** correct

> int FixedFieldFormatter::parsecorrect
> __out == 0 || __out == -1

83. **Filename:** ball_recordjsonformatter.cpp, **Function:** const bsl::string& AttributeFormatter::key(), **Classification:** correct strongest

> const bsl::string& AttributeFormatter::key()correct strongest
> &__out == &d_key

84. **Filename:** ball_recordjsonformatter.cpp, **Function:** RecordJsonFormatter_FieldFormatter * DatumParser::make, **Classification:** correct

> RecordJsonFormatter_FieldFormatter * DatumParser::makecorrect
> __out != nullptr

85. **Filename:** ball_recordjsonformatter.cpp, **Function:** int RecordJsonFormatter::setFormat, **Classification:** trivial

> int RecordJsonFormatter::setFormattrivial
> __out == -1 || __out == 0 || __out != 0

86. **Filename:** ball_rule.cpp, **Function:** int Rule::hash, **Classification:** correct

> int Rule::hashcorrect
> (rule.d_hashValue >= 0 && rule.d_hashValue < size)
> && (rule.d_hashValue == __out && rule.d_hashSize == size)

87. **Filename:** ball_rule.cpp, **Function:** Rule& Rule::operator=, **Classification:** correct strongest

> Rule& Rule::operator=correct strongest
> &__out == this && __out.d_pattern == rhs.d_pattern && __out.d_thresholds == rhs.d_thresholds
> && __out.d_attributeSet == rhs.d_attributeSet && __out.d_hashValue == rhs.d_hashValue
> && __out.d_hashSize == rhs.d_hashSize

88. **Filename:** ball_rule.cpp, **Function:** bsl::ostream& Rule::print, **Classification:** correct

> bsl::ostream& Rule::printcorrect
> &__out == &stream

89. **Filename:** ball_ruleset.cpp, **Function:** int RuleSet::addRule, **Classification:** correct

> int RuleSet::addRulecorrect
> __out == -1 || __out == -2 || __out >= 0

90. **Filename:** ball_ruleset.cpp, **Function:** int RuleSet::ruleId, **Classification:** correct

> int RuleSet::ruleIdcorrect
> __out == -1 || __out >= 0

91. **Filename:** ball_ruleset.cpp, **Function:** bool ball::operator==, **Classification:** correct strongest

```
bool ball::operator==correct strongest
(lhs.numRules() != rhs.numRules() == !__out) && (lhs.numRules() ==
rhs.numRules()
&& std::all_of(lhs.begin(), lhs.end(), [&](const Rule& r)return rhs.ruleId(r)>= 0;)
== __out)
```

92. **Filename:** ball_scopedattribute.cpp, **Function:** bsl::ostream& ScopedAttribute_Container::print, **Classification:** correct

```
bsl::ostream& ScopedAttribute_Container::printcorrect
&__out == &stream
```

93. **Filename:** ball_severity.cpp, **Function:** int Severity::fromAscii, **Classification:** correct

```
int Severity::fromAsciicorrect
__out == 0 || __out == -1
```

94. **Filename:** ball_severityutil.cpp, **Function:** int SeverityUtil::fromAsciiCaseless, **Classification:** correct

```
int SeverityUtil::fromAsciiCaselescorrect
__out == BALL_SUCCESS || __out == BALL_FAILURE
```

95. **Filename:** ball_thresholdaggregate.cpp, **Function:** int ThresholdAggregate::hash, **Classification:** correct

```
int ThresholdAggregate::hashcorrect
__out>= 0 && __out < size
```

96. **Filename:** ball_thresholdaggregate.cpp, **Function:** ThresholdAggregate& ThresholdAggregate::operator=, **Classification:** correct strongest

```
ThresholdAggregate& ThresholdAggregate::operator=correct strongest
(d_recordLevel == rhs.d_recordLevel) && (d_passLevel == rhs.d_passLevel)
&& (d_triggerLevel == rhs.d_triggerLevel) && (d_triggerAllLevel ==
rhs.d_triggerAllLevel)
&& (&__out == this)
```

97. **Filename:** ball_thresholdaggregate.cpp, **Function:** bsl::ostream& ThresholdAggregate::print, **Classification:** correct

```
bsl::ostream& ThresholdAggregate::printcorrect
&__out == &stream
```

98. **Filename:** ball_transmission.cpp, **Function:** const char *Transmission::toAscii, **Classification:** correct

```
const char *Transmission::toAsciicorrect
__out == "PASSTHROUGH" || __out == "TRIGGER" || __out == "TRIGGER_ALL"
|| __out == "MANUAL_PUBLISH" || __out == "MANUAL_PUBLISH_ALL" ||
__out == "(* UNKNOWN *)"
```

99. **Filename:** ball_userfields.cpp, **Function:** bsl::ostream& UserFields::print, **Classification:** correct

> bsl::ostream& UserFields::printcorrect
> &␣␣out == &stream

