# OpenReview forum: "Fun2spec: Code Contract Synthesis At Scale"
_ICLR.cc/2026/Conference — ICLR 2026 Conference Withdrawn Submission_

### Official Review · Reviewer_xv9E · 2025-10-31

**Soundness:** 3
**Presentation:** 2
**Contribution:** 2
**Rating:** 4
**Confidence:** 3

**Summary:**

This paper proposes a LLM-based workflow, Fun2Spec, for synthesizing code contract. The workflow is a loop consisting of a code miner, an LLM-based generator, a specification parser, and a tester. The paper argues the strength of the tool by comparing to the SOTA on other benchmarks. The paper also shows that the tool is industrial-level by applying the tool on large C++ projects.

**Strengths:**

- The paper builds a simple yet effective framework combining specific insights into synthesizing specifications.
- Fun2Spec is evaluated on both small benchmarks and large industrial projects.
- The workflow of Fun2Spec is well presented and explained.

**Weaknesses:**

- The paper has multiple presentation issues and seems rushed (for example, algorithm 1 has overlap to the main text)
- The workflow is not evaluated on (or compared with) any flagship models despite not requiring any finetuning effort
- The paper does not provide artifacts of any sorts.

**Questions:**

- Why is the tool targeting only C++ and not other important languages for verification (like Lean, Verus, etc.)?
- How does the workflow compare with flagship models/agents? For example, the latest GPT/Claude models.

---

### Official Review · Reviewer_vNa2 · 2025-10-31

**Soundness:** 2
**Presentation:** 1
**Contribution:** 2
**Rating:** 2
**Confidence:** 3

**Summary:**

FUN2SPEC is an industrial-strength framework for automatically synthesizing formal specifications (postconditions) in first-order logic for large-scale C++ codebases using large language models. The system employs a five-stage pipeline—mining code repositories to extract functions and contextual information, generating candidate postconditions via LLM prompting with chain-of-thought reasoning, validating and translating specifications into executable C++ assertions, testing them against unit tests, and refining them through error-driven feedback loops—to achieve 85% test validity and generate the strongest postcondition 60% of the time. Evaluated on industrial repositories like BDE and BlazingMQ containing millions of lines of code, FUN2SPEC substantially outperforms the prior NL2POST baseline by 20-35 percentage points across multiple benchmarks while successfully handling complex first-order logic constructs, including quantifiers (FORALL/EXISTS) that enable specification of properties over collections and ranges of values.

**Strengths:**

1. Fun2Spec generates the first-order logic expressions based on the context, and creates assertions from these logic expressions to generate more feedback for refinement.  These logic expressions and assertion-based feedback can examine the designed behavior and provide more grounded and reliable feedback.
2. Compared with the baseline, Fun2Spec shows better performance with different backbone LLMs on HumanEval and FormalSpecCPP benchmark.
3. Human evaluation shows that on complex C++ repos such as BDE and BlazingMQ, the specifications generated by Fun2Spec only have 2% incorrect solutions.

**Weaknesses:**

1. Insufficient experiments to verify their main claims
	1. The average number of code lines in HumanEval-CPP and FormalSpecCPP is very small
	2. For large repos like BDE and BlazingMQ, they only present the Fun2Spec results, without any comparison with other baselines. These results cannot show that the proposed method is better than the others.
	3. Simple baselines such as direct prompting for specification, few-shot prompting for specification, and self-correct prompting should be applied to verify the effectiveness of the proposed pipeline.
	4. The ablation study only considers the impact of the few-shot examples. The effectiveness of each step should also be verified.
2. A lot of formatting issues need to be fixed:
	1. In Line 250-253. There are overlaps between the main context and the algorithm.
	2. Table 2 exceeds the page boundary
	3. Figure 3 should be Table 4, and this figure is never mentioned in the main body.
3. The discussion on qualitative analysis is not enough. While Figure 3 shows that the incorrect cases of Fun2Spec are only 2, it is unclear how the baseline performs. The lack of comparison cannot show the advantage of Fun2Spec.

**Questions:**

1. Can this method extend to other languages?
2. Adding one conclusion section may be better.

---

### Official Review · Reviewer_c23M · 2025-11-01

**Soundness:** 2
**Presentation:** 2
**Contribution:** 2
**Rating:** 6
**Confidence:** 3

**Summary:**

This paper presents Fun2spec, an LLM-based framework for automatically synthesizing formal software specifications (postconditions) for large-scale C++ codebases. The system uses a five-stage pipeline involving code mining, LLM-guided specification generation, parsing, validation through test oracles, and iterative refinement via error feedback. The authors evaluate their approach on benchmark datasets (HumanEval-CPP, FormalSpecCPP) and two industrial C++ repositories (BDE and BlazingMQ), demonstrating 20-35% improvements in test validity over the baseline NL2POST method.

**Strengths:**

1. The work addresses a genuine pain point in software engineering - the lack of formal specifications in existing large codebases. Application to million-line C++ production codebases represents a significant empirical contribution.
2. The paper includes both quantitative metrics (test validity rates) and qualitative analysis (manual inspection of correctness and strength), tested across multiple LLM models and benchmarks.
3. Consistent improvements of 20-35% over state-of-the-art baselines across different models and datasets demonstrate the method's effectiveness.

**Weaknesses:**

1. The core technique (reprompting/self-refinement with error feedback) is well-established in LLM literature and can be found in lots of LLM coding literature.

2. Who verifies the verifier? Using unit tests as a proxy brings about questions such as what happens if the tests are not high-coverage or wrong. This undermines the usefulness of the method.

**Questions:**

N/A

---

### Official Review · Reviewer_FzDY · 2025-11-02

**Soundness:** 2
**Presentation:** 2
**Contribution:** 2
**Rating:** 4
**Confidence:** 4

**Summary:**

This paper presents Fun2spec, a tool for synthesizing formal specifications in first-order logic for C++ functions using LLMs. Fun2spec consists of three main phases: code miner to extract contextual information, LLM generator to synthesis postconditions, and specification tester to filter out invalid candidate specifications. The experiment results demonstrate notable improvements compared with existing approach.

**Strengths:**

- Practical value. Fun2spec targets real, large-scale industrial C++ codebases, which is a step beyond prior works that focus on small or toy examples.
- Well-designed architecture.  The architecture of Fun2spec is thoughtfully designed, and consists of three main phases. The tester phase can help validate the synthesised specifications.
- Evaluation on industrial codebases. Fun2spec is evaluated on industrial codebases (BDE and BlazingMQ) that contain millions lines of code.

**Weaknesses:**

- Evaluation relies on unit tests. The paper’s primary validity metric is unit-test-based (Sections 4.1, 4.2). but there is limited discussion on the completeness of these unit tests. The reliance on tests undermines claims of true behavioral correctness, as tests can miss coverage for program behaviors. Although Section 4.3 evaluates the correctness of specification synthesis by Fun2spec, the metric is vague. For example, there is no clear definition of what is the "strongest correct postcondition".
- Some details are not clear.
  - In Section 4 (Evaluation), it is not specified how many postconditions are generated by Fun2spec and nl2post. The authors should report the exact numbers for transparency and comparability.
  - In Figure 3, it is unclear why Fun2spec generates propositional postconditions, since as stated in line 107, the system only uses first-order logic to represent specifications.
  - In Section 4.3, how many postconditions are evaluated by the authors?

Minor issues:
- The main content and Algorithm 1 appear to overlap.
- In Table 3, some metric titles (e.g., Trivial, Formatting Error) are inconsistent with those used in the main text (e.g., Nontrivial, Invalid Formatting). Please ensure consistency between the table and the text for better readability

**Questions:**

- Could the authors clarify how “the strongest correct postcondition” is formally defined in Section 4.3?
- Can the authors provide detailed statistics regarding unit test coverage, and discuss how coverage limitations affect confidence in the correctness of the generated specifications.
- I noticed that the paper of nl2postcond proposes the metrics of test-set correctness and test-set completeness for code mutants to evaluate the postcondition, why did the authors not adopt these metrics in the evaluation? Moreover, the paper nl2postcond reports an evaluation on industrial-scale projects and claims to have detected 64 historical bugs. How does Fun2spec perform in terms of bug detection? If no similar experiment was conducted, could the authors explain the rationale for omitting such an evaluation?

**Details Of Ethics Concerns:**

N.A.

---

### Note · Authors · 2025-11-16

**Comment:**

We thank the reviewers for the constructive feedback. We will address these points in a future submission

**Withdrawal Confirmation:**

I have read and agree with the venue's withdrawal policy on behalf of myself and my co-authors.